# Lateral dispersion is required for circuit integration of newly generated dentate granule cells

Jia Wang[1,2,7], Jia Shen [1,3,7], Gregory W. Kirschen[1,4,7], Yan Gu [1,5], Sebastian Jessberger[6] & Shaoyu Ge[1]

The process of circuit integration of newly-generated dentate granule cells of the hippocampus has been presumed to be a dynamic process. In fact, little is known regarding the initial development of newly generated neurons prior to circuit integration and the significance of this stage for circuit integration. Here, using advanced live imaging methods, we systematically analyze the dynamic dispersion of newly generated neurons in the neurogenic zone and observe that cells that are physically adjacent coordinate their lateral dispersion. Whole-cell recordings of adjacent newly generated neurons reveal that they are coupled via gap junctions. The dispersion of newly generated cells in the neurogenic zone is restricted when this coupling is disrupted, which severely impairs their subsequent integration into the hippocampal circuit. The results of this study reveal that the dynamic dispersion of newly generated dentate granule cells in the neurogenic zone is a required developmental stage for circuit integration.

[1] Department of Neurobiology and Behavior, SUNY at Stony Brook, Stony Brook, NY 11794, USA. [2] Biomedical Pioneering Innovation Center, Peking University, 100871 Beijing, China. [3] Genetics Graduate Program, SUNY at Stony Brook, Stony Brook, NY 11794, USA. [4] Medical Scientist Training Program, SUNY at Stony Brook, Stony Brook, NY 11794, USA. [5] Center for Stem Cell and Regenerative Medicine, Zhejiang University School of Medicine, 310058 Hangzhou, China. [6] Laboratory of Neural Plasticity, Faculties of Medicine and Science, Brain Research Institute, University of Zurich, 8057 Zurich, Switzerland. [7]These authors contributed equally: Jia Wang, Jia Shen, Gregory W. Kirschen. Correspondence and requests for materials should be addressed to S.G. (email: shaoyu.ge@stonybrook.edu)

Dentate granule cells (DGCs) are continuously generated in the hippocampus of the adult brain[1–3]. Over the past decade, we and others have extensively characterized the generation of DGCs, the formation of their synapses, and their physiological functions once integrated into the granule cell layer (GCL)[4–9]. However, the initial development of DGCs in the neurogenic zone and how this influences their subsequent integration into the circuit remain largely unknown.

New DGCs are generated in the subgranular zone (SGZ)[1,2], a neurogenic zone between the GCL and the hilus. These cells can be marked by 5-bromo-2′-deoxyuridine and cell-type-specific markers or by transgenic methods and observed to disperse laterally along this zone[10–12]. However, the dynamics and mechanisms of this dispersion have remained elusive, because of the inability to precisely birth date new cells and monitor their development in real time. A cluster of studies have shown that dispersion in the ventricular zone is required for cortical neural circuit formation in embryonic brain development[13,14]. The physiological relevance of this dispersion of newborn DGCs for subsequent circuit integration remains unknown.

In this study, we use novel live imaging methods to analyze the migration of newly generated DGCs in the neurogenic zones of freely moving animals. We discover the coordinated dispersion of adjacent cells. These neighboring newly generated DGCs are found to be coupled via gap junctions, which is required for proper lateral dispersion and subsequent circuit integration.

## Results

**Live imaging of the dispersion of newly generated DGCs.** New DGCs are continuously generated in the neurogenic zone beneath the pre-existing DGC layer in the adult brain[1–3]. Analyses of genetically and chemically labeled new DGCs revealed that they disperse laterally in the neurogenic zone prior to synapse formation[11,12]. However, the dispersion kinetics and physiological role of the dispersion for subsequent circuit integration remain unknown.

To analyze the kinetics of this dispersion in the present study, newly generated DGCs were birth-dated via a retrovirus encoding green fluorescent protein (GFP) as previously described[4]. Five days post infection (dpi), when retrovirus-labeled cells have become postmitotic with DGC features[4] (Supplementary Fig. 1), we observed that most GFP+ DGCs were horizontally positioned relative to the axis of the GCL (Fig. 1a). More importantly, most of these neurons showed a typical leading process with proximal cytoplasmic bulging (spindle expansion) along the neurogenic zone (Supplementary Fig. 2), suggesting that these cells were migrating[15,16]. To characterize the period during which these cells disperse along the neurogenic zone, we measured the angles of the cellular axes of GFP+ DGCs at 5, 6, and 7 dpi relative to the axis of the GCL. We found that during this period, the percentage of GFP+ DGCs between the angles of −10° and 10° decreased sharply (Fig. 1b), suggesting that horizontal dispersion along the neurogenic zone likely occurs between 5 and 7 dpi. After this time window, newly generated DGCs likely begin a radial migration (i.e., perpendicular to the neurogenic zone)[17,18].

To assess the dynamics of the dispersion of newly generated DGCs along the neurogenic zone, we utilized our newly established in vivo imaging method to monitor retrovirus-labeled DGCs in freely moving mice. As illustrated in Fig. 1c, 2 days after the infusion of the GFP retrovirus, the tip of a gradient-index (GRIN) lens was positioned directly above the DGC layer at a location distal to the injection site, with the focal plane at the subgranular zone. 3 days after recovery, GFP+ cells were imaged for 2 days with a miniature endoscope (Inscopix, Inc., Palo Alto, CA) in freely moving mice, as recently described[19]. The majority of GFP+ cells dispersed extensively between 5 and 6 dpi (Supplementary Movie 1). Most cells dispersed in one direction, but a small portion of them dispersed back and forth. After imaging, the positioning of the lens was verified as described previously[19]. The dispersion speed, net displacement (the distance between the initial and final positions), and total distance traveled of migrating cells with a polarized morphology and typical leading processes were measured as shown in Fig. 1d. As the endoscope has a single focal plane in the dorsoventral axis (~290 μm working distance in this brain tissue)[19,20] and only cells that remained in the focal plane for the duration of the analysis were examined, these cells were considered to have dispersed laterally in the same rostrocaudal-mediolateral plane they were originally detected. As presented in Fig. 1e, the average displacement over the 10 h of imaging was ~64.8 μm, consistent with measurements in fixed tissue of transgenically labeled new DGCs[11]. The average total distance traveled during the capture time window was ~300.3 μm with an average speed of ~5 μm/h, indicating a highly dynamic process during this period. We further verified this in vivo analytic method did not affect the integrity of the neurogenic niche by analyzing DCX+ cells, showing similar number and processes of DCX+ cells (Supplementary Fig. 3). This suggests that this imaging method is adequate for analyzing natural occurring of newborn neurons. Overall, approximately half of the detected GFP+ cells dispersed laterally, and the other half did not disperse or disappeared during imaging, likely having undergone either apoptosis or radial migration in the dorsoventral axis.

The dispersion of GFP+ cells was dynamic, and adjacent cells were often observed in possibly coordinated migration with an alternation of the lead position (Supplementary Movie 2), akin to a game of leapfrog. Note that a recent study showed some newborn DGCs in the neurogenic zone contacted microvessels[11]. It has been postulated that newborn DGCs may disperse along the microvessel. Interestingly, among 11 analyzed cells, we indeed observed a couple of interesting cells, as shown in Supplementary Movie 3, wrapped and migrated along a vessel. This suggests the possibility of the vessel-based migration mechanism. However, we did not observe clear association with microvessel for most newborn DGCs. We thus suspected that the 'leapfrog' dispersion might be another important mechanism by which newly generated DGCs disperse in the neurogenic zone. To examine this observation further, adult brain sections were cultured for 4 days after newly generated DGCs were virally labeled via a previously described method[21]. Newborn DGCs were imaged every 30 min for 3 days with a high-throughput live imaging system comprising a spinning disk confocal microscope and incubator (Fig. 1f). The imaging revealed that approximately half of the GFP+ cells dispersed horizontally along the neurogenic zone, similar to that observed in vivo. The other half died during imaging. Interestingly, ~10% of the entire GFP+ cells exhibiting a typical DGC morphology migrated radially into the dentate GCL. The speed and distance of the laterally dispersing cells were measured (Fig. 1g, h), and the results were consistent with the measurements from freely moving animals (Fig. 1e). As shown in Fig. 1g, h and Supplementary Movies 2 and 4, 6/17 GFP+ cell clusters (clusters of 3–5 physically adjacent cells) indeed exhibited leapfrog-like dispersion. This proportion may be an underestimate, as only a single layer of sectioned hippocampal tissue was imaged.

Altogether, the in vivo and in vitro live imaging of retroviral birth-dated and labeled DGCs revealed that newly generated DGCs disperse in the neurogenic zone between 5 and 7 dpi. Moreover, we discovered that the dispersion of physically adjacent cells occurs in a coordinated leapfrog manner, potentially representing the unit for chain dispersion as

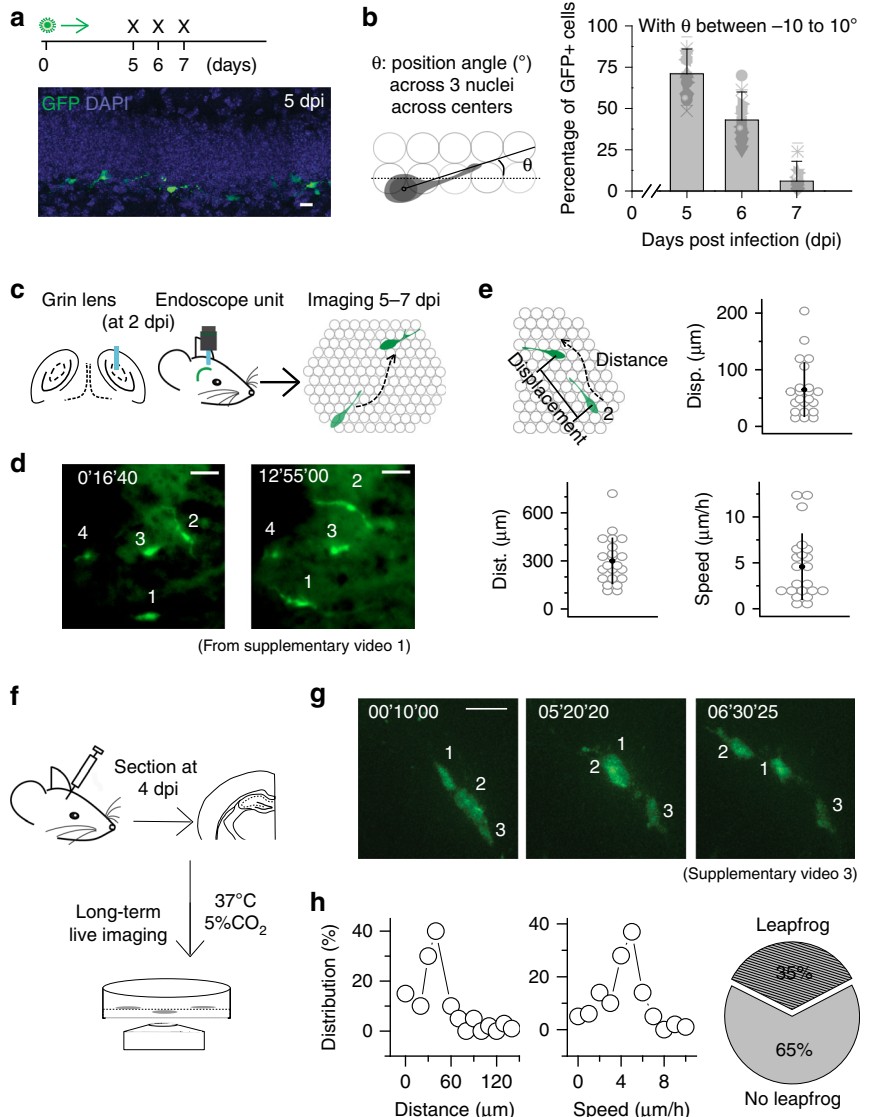

**Fig. 1** Dynamic integration of newly generated hippocampal neurons in the neurogenic zone. **a** Experimental timeline of retroviral birth dating (top) and representative image of retrovirally birth-dated GFP+ newborn DGCs at 5 dpi (bottom). Scale bar is 15 μm. **b** Schematic of the measured angle of the leading process of a given DGC with respect to the SGZ (left). Percentages of GFP+ newborn DGCs with angles of the leading processes between −10° and 10° over time (right, $n = 3$–4 mice). **c** Diagram of in vivo deep brain imaging of newborn DGCs in the SGZ. **d** Representative images of migrating cells over time in vivo, taken from Supplementary Movie 1. Scale bar is 50 μm. **e** Schematic of dispersion (top left) and measurements of absolute distance, displacement (final−initial), and speed of imaged GFP+ DGCs during the recorded time window ($n = 3$ mice). **f** Diagram of adult hippocampal slice culture imaging of newborn DGCs in the SGZ. **g** Representative images of migrating cells in adult hippocampal slices culture over time, taken from Supplementary Movie 3. Scale bar is 20 μm. **h** Distributions of migration (absolute) distance and average speed of newborn DGCs in the SGZ and the percentages of leapfrog-like dispersing cells from all the observed newborn cells in adult hippocampal slice culture ($n = 3$ mice)

previously observed in the rostral migratory stream[22]. We next aimed to determine the physiological role of this dispersion toward the subsequent circuit integration of the newly generated DGCs.

**Electrical coupling between newly generated DGCs**. The physical interaction between newly generated DGCs during dispersion in the neurogenic zone led us to speculate that these cells might be electrically coupled, as observed for radially migrating cortical neurons in the embryo[23,24]. Therefore, whole-cell recordings were performed of physically adjacent GFP+ cells in acutely prepared 270-μm-thick hippocampal slices at 5 dpi, as illustrated in Fig. 2a. A brief current was injected into one of the paired cells, and the response with a slightly smaller current from

the other cell was recorded (and vice versa) as previously described[25]. Two cells that exhibited reciprocal responses to stimulation were defined as electrically coupled. As shown in Fig. 2b, c, 3/14 pairs were coupled, similar to the pairing ratio of migrating cells in cortical embryonic sections[24]. However, only 1/13 pairs were coupled at 7 dpi, suggesting that the cells uncouple upon the completion of lateral dispersion (Fig. 1b). As these results are from brain sections, higher percentages of coupled cells are expected in vivo, as predicted in embryonic recordings[24]. The observed electrophysiological coupling was blocked during bath application of a generic connexin blocker, carbenoxolone (100 μM, Fig. 2d), demonstrating that the coupling was via gap junctions.

To more efficiently analyze the lateral dispersion, we utilized a Cre-Loxp-based two-virus system to sparsely label individual

 **3**

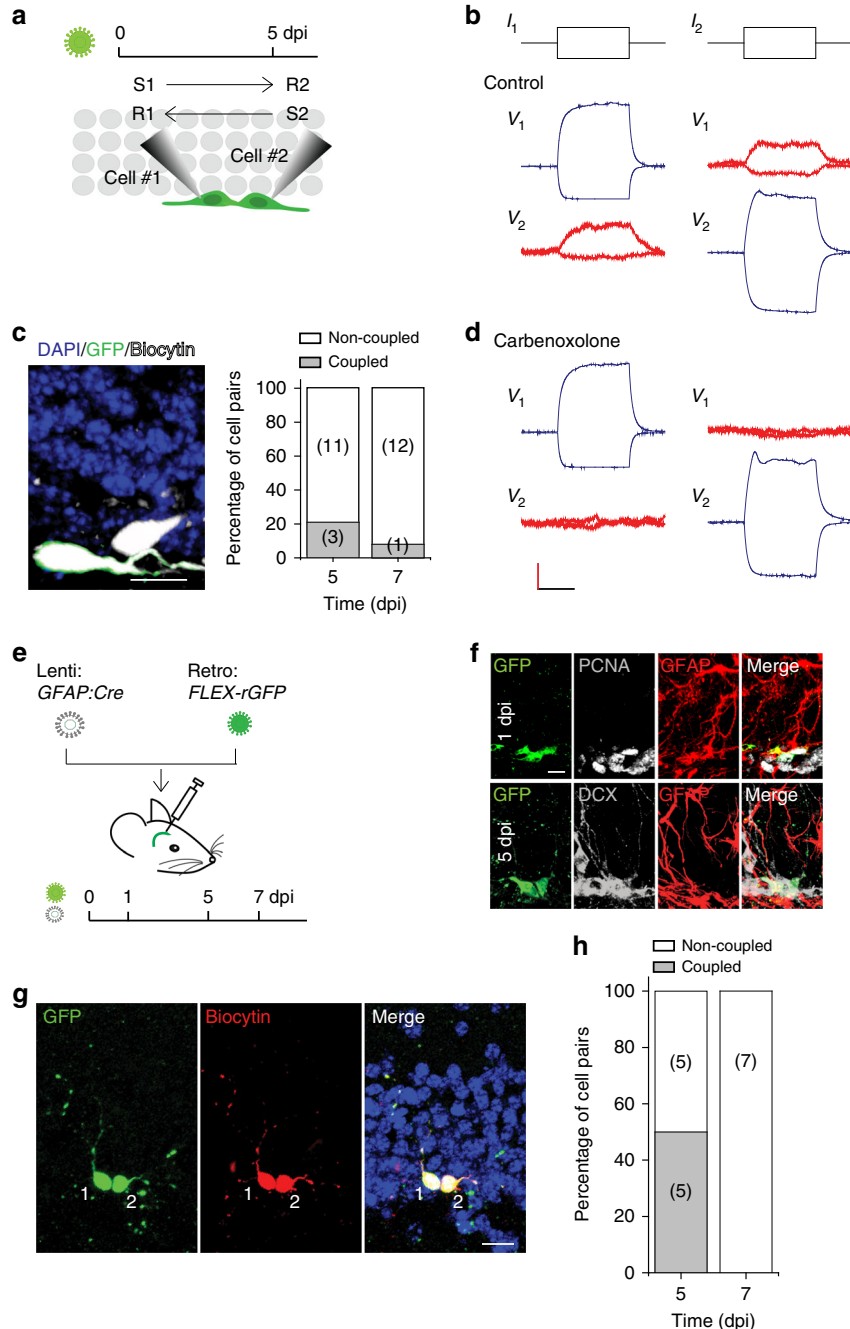

**Fig. 2** Gap junction coupling between dispersing newly generated DGCs. **a** Schematic of the experimental setup, in which retrovirally labeled GFP-expressing newborn DGCs are whole-cell patch clamped and examined for reciprocal electrical coupling. **b**, **d** Representative current and voltage traces taken from two adjacent, electrically coupled, patched newborn DGCs at 5 dpi without **b** or with **d** 100 μM carbenoxolone treatment. The vertical scale bars are 100 pA/10 mV. Horizontal scale bar is 100 ms. **c** Representative image of biocytin-filled, retrovirally birth-dated 5-day-old DGCs from the experiment shown in panels **b** and **d** (left). Percentages of pairs of cells that were reciprocally electrically coupled (right). Scale bar is 10 μm ($n = 4$ mice). **e** Diagram and experimental timeline of the two-virus method of sparsely labeling clusters of newborn DGCs, in which lenti-*GFAP-Cre* and the FLEX-reverse GFP retrovirus were co-injected into the dentate gyrus. **f** Example of two cell clusters of GFP$^+$ cells in the SGZ of the dentate gyrus using the two-virus method at 1 and 5 dpi, respectively. These sections have been stained with PCNA and GFAP for 1 dpi, and DCX and GFAP for 5 dpi. Scale bar is 10 μm. **g** Representative images showing biocytin-filled, retrovirally birth-dated 5-day-old DGCs from the experiment in **e**. Scale bar is 15 μm. **h** The number of pairs of cells that were reciprocally electrically coupled or non-coupled from the experiment in **e** ($n = 3$ mice)

clusters of newly generated DGCs and measured their dispersion[19,26,27]. Briefly, we generated Cre-dependent retroviral vectors for the expression of GFP and dnCX43 (FLEX-rGFP and FLEX-rdnCX43-GFP, respectively), in which the transgene sequence was reversed. High titers of these retroviruses along with lentiviruses carrying the *Cre* gene under the *GFAP* promoter,

active in the progenitors[10] (lenti-*GFAP-Cre*), were injected into the dentate gyrus regions in mice to target newly generated DGCs (Fig. 2e). We further confirmed FLEX-rGFP-infected newborn DGCs in these clusters at 5 dpi were immature DGCs by immunostaining as shown in Fig. 2f. We next analyzed the coupling efficiency of newborn DGCs in labeled cell clusters.

We recorded newborn DGCs at 5 dpi (Fig. 2g). Interestingly, we recorded 10 pairs at 5 dpi from three mice and found five pairs were electrically coupled (Fig. 2h). Consistent with the observation in Fig. 2c, at 7 dpi, we recorded seven pairs in cell clusters and found no pair was coupled (Fig. 2h). This set of experiments suggests that newborn DGCs from a single progenitor may be preferentially coupled.

**Electrical coupling is required for the lateral dispersion**. To assess the importance of electrical coupling via gap junctions for lateral dispersion, we examined the neurogenic zone migration of cells deficient for connexin 43 (CX43), a gap junction channel protein that is highly expressed in neural stem cells and their newborn progeny in the dentate gyrus[28] (Supplementary Fig. 4). For these experiments, CX43-floxed animals[29] were injected with a Cre-GFP retrovirus, and GFP$^+$ cells were imaged at 5 dpi using our in vivo imaging method (Fig. 3a). The imaging results revealed that the dispersion of CX43-depleted GFP$^+$ cells within the neurogenic zone was substantially reduced (Fig. 3b and Supplementary Movie 5).

As gap junctions mediate both electrical coupling and cell–cell adhesion, we next assessed whether the electrical coupling and not cell adhesion is important for coordinating the lateral dispersion of newly generated DGCs by using a dominant negative (dn) CX43 with dysfunctional coupling but intact adhesion capability[23]. To more efficiently analyze the lateral dispersion, we utilized the Cre-Loxp-based two-virus system as described in Fig. 2e using a different timeline (Fig. 3c) to sparsely label individual clusters of newly generated DGCs and measured their dispersion[19,26,27]. At 5 dpi, ~11–17 clusters of GFP$^+$ newly generated DGCs (2–5 cells per cluster) were observed per animal (Fig. 3d). Rarely was more than one cluster observed in a single 120-μm hippocampal section, indicating the feasibility of this method for measuring the relative distances between cells in a cluster to study their lateral dispersion, as we recently demonstrated using statistical modeling[30]. Importantly, the fluorescent signals of labeled cells were sufficiently robust for electrophysiological and morphological analyses without any staining for signal amplification.

In slices from animals infected with the FLEX-rGFP virus, the distances between the nearest GFP$^+$ cells in clusters increased from 5 to 7 dpi, with little further increase from 7 to 8 dpi (Fig. 3e, f), similar to the time course of horizontal positioning observed in vivo (Fig. 1b). To exclude the possibility that the increase in distance was a result of cell loss, the numbers of cells per cluster were counted 5, 6, and 7 dpi. We found that the numbers were stable at ~3.5 cells/cluster across the three time points, suggesting that there was no detectable cell loss during the lateral dispersion period. By contrast, the distances between dnCX43-GFP$^+$ cells in the clusters analyzed did not change between 5 and 7 dpi, suggesting a defect in lateral dispersion (Fig. 3g, h). The number of dnCX43-GFP$^+$ cells per cluster was between 3 and 5 (3.2 cells on average), indicating that the CX43 deficiency did not alter the survival of newly generated DGCs at the time points examined.

Altogether, these results reveal that gap junction coupling preferentially occurs between newly generated DGCs derived from a single progenitor and that the coupling is necessary for lateral dispersion in the neurogenic zone.

**Lateral dispersion is required for circuit integration**. We next sought to determine the role of lateral dispersion of newly generated DGCs in their circuit integration into pre-existing neural network during the first 2 weeks after their birth[4,6,8,9]. First, the number and length of dendrites were measured in newly generated DGCs expressing dnCX43 at 7 and 14 dpi, as previously

described[4] (Fig. 4a). Consistent with previous observations[4,8,18], the branch numbers and total dendritic lengths of GFP$^+$ cells increased from 7 to 14 dpi, albeit to a lesser extent than in control cells expressing the FLEX-rGFP virus (Fig. 4b). To determine whether the altered morphologies resulted from delayed development, the membrane properties of both dnCX43$^+$ and control GFP$^+$ DGCs were recorded at 5, 7, and 14 dpi. As presented in Fig. 4c, there was no obvious difference in intrinsic membrane properties between these two groups. Moreover, the radial migration of these newly generated DGCs was not different at 14 dpi (Supplementary Fig. 5), suggesting the dispersion into the dentate GCL was not impacted by dnCX43 expression.

To further confirm the defect in integration was from disrupted lateral dispersion avoiding potential confound effects from defective proliferation and differentiation of neural stem cells, we conditionally expressed the dnCX43 in new DGCs of adult mice via a retroviral vector in which the transgene is only expressed in the presence of doxycycline, as previously shown[17,31]. We induced dnCX43 expression in newborn DGCs at 4 dpi and analyzed the total branch numbers and dendritic lengths at 14 dpi. Consistent with the findings shown in Fig. 4b, the induction of dnCX43 expression resulted in substantially stunted dendritic growth compared to that of the cells expressing only GFP at these time points (Fig. 4d). When we induced the expression of dnCX43 at 7 dpi, at which time lateral dispersion is largely finished (Figs. 1 and 3), we found that the measurements of dendrites (i.e., total branch number and dendritic length) from these cells at 14 dpi were comparable to those from controls (Fig. 4e).

We then measured functional synapse formation onto newborn DGCs by assessing the profile of spontaneous or evoked synaptic transmission. In cohorts of animals, we induced the expression of GFP only or dnCX43 at 4 dpi and performed experiments at 21 dpi, at which time points newborn DGCs are well functionally integrated[4]. As expected, although these dnCX43$^+$ cells developed normal intrinsic membrane properties (Fig. 4f), they had severe morphological defects (Fig. 4g, h), which have been further confirmed by depleting CX43 (Supplementary Fig. 6). We performed whole cell recording in control and dnCX43$^+$ newborn DGCs at 21 dpi, while electrically stimulating the outer two-third of the molecular layer of the dentate gyrus to activate entorhinal projections as previously described[31]. We analyzed eEPSCs in the presence of 5 μM bicuculline. Successful synaptic transmission was recorded from 100% of control newborn DGCs but from only $65 \pm 11.2\%$ of dnCX43$^+$ DGCs. The mean peak amplitude of eEPSCs in responsive dnCX43$^+$ DGCs was smaller than controls (Fig. 4i). The frequency of spontaneous glutamatergic synaptic currents (sEPSCs) of control newborn DGCs was $1.04 \pm 0.29$ Hz as compared to $0.48 \pm 0.19$ Hz in dnCX43$^+$ DGCs. There was no significant difference in sEPSCs mean amplitude (Fig. 4j).

Altogether, these results revealed that the lateral dispersion of newly generated DGCs in the neurogenic zone is required for subsequent circuit integration.

**Discussion**
We used novel live imaging and genetic electrical uncoupling methods to examine the early developmental stages of newborn DGCs in the adult brain. Specifically, we focused on the dispersion kinetics of newly generated DGCs in the neurogenic zone and the role of lateral dispersion in subsequent circuit integration. We discovered stereotypical patterns of dispersion, including a previously unappreciated leapfrog form of migration of physically adjacent cells. We also found that newly generated DGCs are gap junction coupled and that this functional coupling is essential for dispersion in the neurogenic zone. Finally, we showed that the

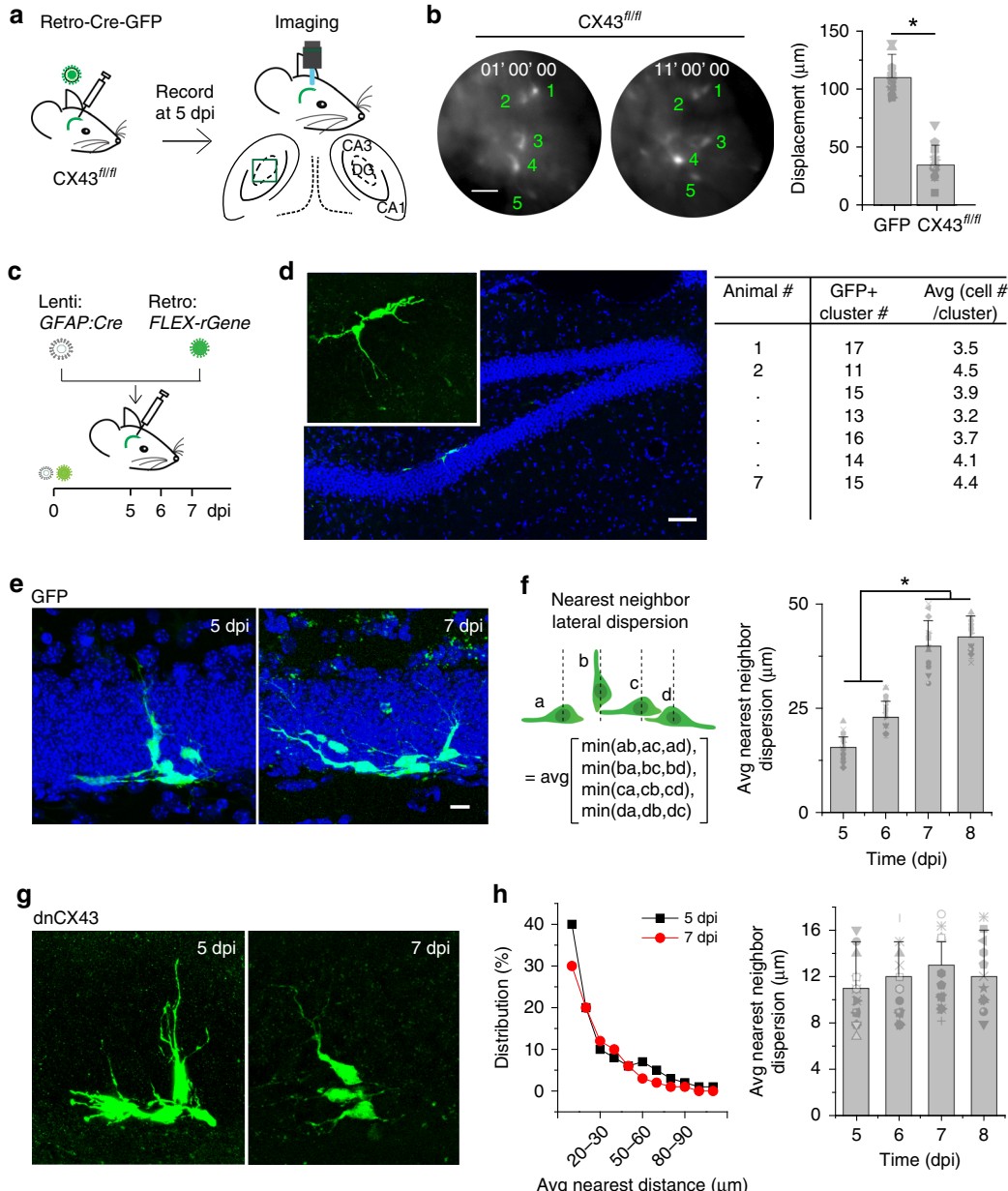

**Fig. 3** Electrical uncoupling of newly generated DGCs disrupts lateral dispersion. **a** Schematic of the system for in vivo imaging of birth-dated newborn DGCs with genetic deletion of CX43 via introduction of Cre recombinase fused to GFP (Cre-GFP) into CX43-floxed animals (CX43 knockout). **b** Representative serial in vivo images from freely moving mice in the CX43 knockout condition from the experiment shown in panel **a** at 1 and 11 h after the beginning of the recording session (left). Scale bar is 20 μm. Plot of the average displacement of cells in the control animals (GFP, shared with Fig. 1e) and CX43 knockout conditions (right). Two-tailed unpaired *t* test, *$P < 0.001$ ($n = 3$ mice). **c** Diagram and experimental timeline of the two-virus method of sparsely labeling clusters of newborn DGCs, in which lenti-*GFAP-Cre* and the FLEX-reverse dnCX43-GFP or reverse GFP retrovirus were co-injected into the dentate gyrus. **d** Example of a cluster of three GFP⁺ cells in the SGZ of the dentate gyrus sparsely labeled using the two-virus method (left) at 5 dpi. The inset is an enlarged view of two right GFP⁺ cells. Cluster numbers and sizes observed in each animal (right). Seven mice had been included in this analysis. Scale bar is 50 μm. **e**, **g** Representative images of sparsely labeled cell clusters from control (GFP only) **e** and dnCX43 **g** conditions at 5 and 7 dpi. Scale bar is μm. **f** Diagram for calculating the average nearest-neighbor lateral dispersion of a sparsely labeled cluster (left). Quantification of the average nearest-neighbor lateral dispersion of cell clusters in control condition across the experimental timeline (right). **h** Distribution (left) and quantification (right) of the average nearest-neighbor lateral dispersion of cell clusters in dnCX43 condition at given time points. ANOVAs were performed for the comparisons in **f** and **h** ($n = 3$–4 mice; *$P < 0.05$).

dispersion of newly generated DGCs in the neurogenic zone is required for subsequent circuit integration.

It has been a challenge to track the dispersion of hippocampal newborn neurons in the adult brain, hindering a clear understanding of the dispersion pattern and underlying mechanisms. Using a live imaging method we newly established, we were able to monitor the dispersion of these cells and observe leapfrog-like

pattern of dispersion. A pattern of chain dispersion of newborn cells in the rostral migratory stream of the adult brain has been well-known for several decades[22]. Leapfrog dispersion would not be conducive for such a mass migration of thousands of stacked cells but may be more convenient than chain migration in the case of the dentate gyrus, where there are fewer newborn cells that have a shorter distance to travel. Although the complex fiber

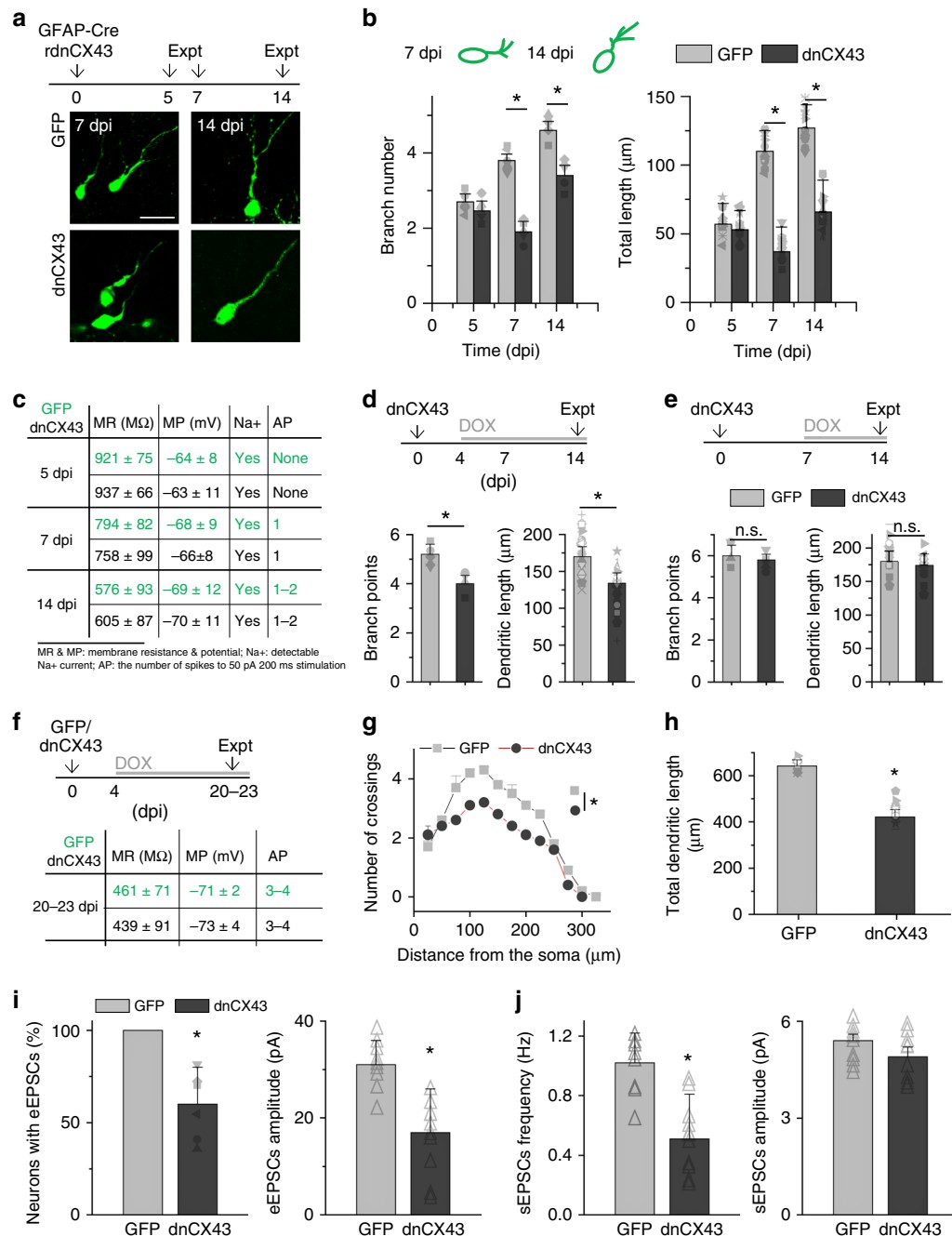

**Fig. 4** Disruption of electrical coupling impairs integration of newly generated DGCs. **a** Experimental timeline of birth-date labeling and introduction of a dnCX43 into newborn DGCs. Representative images of control (GFP) and dnCX43-expressing newborn DGCs at 7 and 14 dpi (bottom). Scale bar is 20 μm. **b** Plots of numbers of apical dendrite branch points (left) and total lengths (right) at 5, 7, and 14 dpi for control and dnCX43-expressing DGCs. Branch points, each data point indicates the average of one animal, two-way ANOVA, 7 dpi, *$P < 0.001$; 14 dpi, *$P < 0.001$. Total length, each data point indicates the total length of one cell, two-way ANOVA, 7 dpi, *$P < 0.001$; 14 dpi, *$P = 0.005$ ($n = 3$–6 mice). **c** Table of electrophysiological properties of control and dnCX43+ cells at 5, 7, and 14 dpi. **d**, **e** Experimental timeline of birth-date labeling and introduction of dnCX43 into newborn DGCs from 4 or 7 dpi, with data collected at 14 dpi (top). Plots of branch numbers and total dendrite lengths at 14 dpi (data points shown similar as in **b**, bottom). Two-tailed unpaired $t$-tests, *$P < 0.05$, n.s. $P > 0.05$ ($n = 3$–5 mice). **f** Table of electrophysiological properties of control and dnCX43+ cells at 20–23 dpi. **g**, **h** In **g**, Sholl analysis of the dendritic tree of control and dnCX43+ at 21 dpi. In **h**, a summary of total dendritic length for control and dnCX43+ cells at 21 dpi ($n = 25$–38 neurons from three mice of each group. For **g**, statistical significance was determined by Student's $t$-test; for **h**, ANOVA, *$P < 0.05$). **i**, **j** Glutamatergic synaptic transmission recorded from control and dnCX43+ DGCs at 21 dpi. **i** The percent of recorded newborn DGCs with detectable synaptic transmission on the left. On the right is eEPSCs amplitude. **j** Spontaneous glutamatergic synaptic transmission, frequency and amplitude ($n = 7$–12 neurons from three animals of each group; ANOVA, *$P < 0.05$). The value from each cell was used for statistical analysis and error bars represent SEM

network of radial glia-like (RGL) cells has been postulated to serve as a guide for migrating cells[32], in this study we had not performed additional experiments to determine this possibility. Nevertheless, these glia-like cells may form an important matrix to restrain the dispersion of newly generated DGCs in the neurogenic zone, similar to the role observed in the rostral migratory stream[22]. Besides this scaffold, the vascular infrastructure may provide a local railway system for neuronal dispersion (Sun et al.[11] and Supplementary Movie 3). However, a large portion of newborn DGCs at 5–8 dpi seem to migrate along the neurogenic zone showing no obvious association with microvessels, raising the possibility the dispersion may employ different mechanisms across the initial development.

A major unanswered question in the field of neurogenesis is how newly generated DGCs coordinate their dispersion in the neurogenic zone. The findings presented here indicate that some of these cells are electrically coupled via gap junctions, which is essential for proper dispersion. Whereas gap junctions are important for neuronal dispersion in the embryonic cerebral cortex[23,24,33], the findings here demonstrate their importance for the lateral dispersion of newly generated DGCs in the adult brain. Electrical coupling is not confined to newly generated DGCs, a recent study has shown that RGL cells are electrically coupled and this coupling plays an important role in regulating proliferation and differentiation of neural stem cells. Interestingly, although both population of cells are coupled with their own lineages, RGL cells and newborn DGCs showed no cross-coupling[28].

Another important finding presented here is the role of DGC dispersion in subsequent circuit integration. The disruption of the dispersion of newborn DGCs in the neurogenic zone resulted in rudimentary dendritic processes highly suggestive of poor integrative capacity, although their tested intrinsic properties were normal. Moreover, the disruption of electrical coupling at a stage when the cells have already dispersed did not affect dendrite formation, suggesting that the altered morphological development was due to defective lateral dispersion induced by the uncoupling at early stages.

The lateral dispersion of newly generated DGCs likely provides them with sufficient separation to successfully compete for circuit integration, as we previously hypothesized[34]. Although we did not extend our tests to later development stages in the present study, the findings strongly suggest a mechanism by which newly generated DGCs disperse to enable them to compete for the opportunity to integrate, providing a means for adult circuits to regulate the addition rate of newborn DGCs. This is also reminiscent of the well-known overshoot and pruning of neurites used by newly integrating DGCs[35]. Altogether, our results present a novel potential mechanism, that the adult brain may use to regulate the homeostatic integration of newly generated DGCs.

## Materials and methods

**Animal experiments**. All surgeries and experimental procedures were approved by the Stony Brook University Animal Use Committee and in accordance with the guidelines of the National Institutes of Health. Experiments were conducted using 6–8-week-old C57BL/6 mice of both sexes (Charles River Laboratories, Wilmington, MA) and B6.129S7-*Gja1tm1Dlg*/J (Cx43 floxed) mice (stock number 008039; The Jackson Laboratory, Bar Harbor, ME). For all surgeries, mice were anesthetized with a ketamine/xylazine cocktail (200 mg/kg, i.p.) and administered buprenorphine HCl (0.05 mg/kg, i.p.) for immediate postsurgical analgesia. Mice were placed on a 37 °C heating pad immediately after surgery for 2 h to recover. For conditional expression of dnCX43 or Cre-GFP, doxycycline was provided in the drinking water (2 mg/mL in a 0.4 M sucrose solution).

**Viruses**. Retro-virus and lentivirus were packaged in our own laboratory and used to deliver genes of interest[5]. Ontogenetic labeling viruses were co-infused (0.5 μL/ injection site) into the dentate gyrus at two sites (stereotaxic coordinates, −2.0 mm from bregma, ±1.6 mm lateral, 2.5 mm in dorsoventral and −3.0 mm from bregma, ± 2.6 mm lateral, 3.2 mm in dorsoventral).

**Slice and in vivo physiology**. Mice were perfused with a cutting solution at 5–8 or 20–23 dpi after retroviral injection. Electrophysiological recordings were performed at 32–34 °C. Electrical stimulation experiments of the entorhinal perforant path used standard bipolar electrodes to determine evoked glutamatergic synaptic transmission in cells held at −65 mV. The stimulus intensity was maintained for all tests. Spontaneous synaptic activity was examined in the presence of 1 μM TTX during 5 min continuous sweeps recorded under voltage-clamp at −65 mV in the presence of 5 μM bicuculline. All chemicals used were purchased from Sigma.

**In vitro slice imaging**. Mice received injections of a GFP retrovirus as described above in the section "Viruses" and were then perfused with chilled (4 °C) slice culture medium (50% Eagle's Base 1 essential medium, 25% Eagle's balanced salt solution, 25% horse serum, 25 mM HEPES-Na, 0.5 mM L-glutamate, 25 mM glucose, 100× penicillin/streptomycin) at 4 dpi. The brains were extracted and sectioned into 100-μm coronal sections and placed in a spinning disk confocal incubator (CV1000, Olympus) at 37 °C with 5% $CO_2$ continuously supplied. Sections were imaged at ×40 magnification via a spinning disc confocal microscope at 10-min intervals.

**In vivo imaging and analysis**. In vivo fluorescence imaging was performed essentially as previously described[19]. Briefly, mice were injected with a GFP retrovirus as described above in Viruses. 2 days later, a lens probe (outer diameter, 1.0 mm; length, 4.0 mm; numerical aperture [NA], 0.5) was implanted 0.2–0.3 mm superior to the SGZ to allow for the working distance of the lens. We typically were able to locate GFP+ cells during lens implantation. After another 2–3-day recovery, in vivo fluorescence imaging was performed with a miniature microscope (Inscopix, Inc.) to track the dispersion of retroviral-labeled GFP+ cells. Five frames were taken every 10 min for a period of ~48 h. The five frames recorded at each time point were combined into a z-axis projection image; the projection images obtained over the 48-h imaging period were then stacked into one video. The movement of the GFP+ cells was analyzed in ImageJ.

**Immunofluorescent staining and confocal imaging**. Mice were deeply anesthetized with urethane (200 μg/kg) and perfused transcardially with phosphate-buffered saline followed by 4% formaldehyde. The brains were removed, fixed overnight in 4% formaldehyde, and then transferred to a 30% (w/v) sucrose solution and stored at 4 °C until sectioning. The brains were sectioned into 80-μm coronal sections to avoid disrupting cell clusters, covering the entire anterior/posterior axis of the DG. Immunohistochemistry was performed by blocking sections in 1% donkey serum in phosphate-buffered saline with 0.025% Triton for 1 h at room temperature and then incubating them overnight with shaking at 4 °C with the following primary antibodies: GFP (rabbit polyclonal antibody, 1:1000; Sigma-Aldrich, St. Louis, MO), Prox1 (mouse monoclonal, 1:500; Millipore, Burlington, MA), PCNA (goat polyclonal, 1:100; Santa Cruz Biotechnology, Dallas, TX), GFAP (rabbit polyclonal, 1:500; Dako, Agilent, Santa Clara, CA), and DCX (goat polyclonal antibody, 1:1000; Santa Cruz Biotechnology). The sections were then incubated for 3 h with shaking at room temperature with the following secondary antibodies: Alexa 488-conjugated donkey anti-rat antibody (1:1000; Abcam), Alexa 594-conjugated donkey anti-rat antibody (1:1000; Abcam), Alexa 488-conjugated donkey anti-rabbit antibody (1:1000; The Jackson Laboratory), and Alexa 488-conjugated donkey anti-goat antibody (1:1000; Abcam).

Images were obtained on an Olympus FLV1000 confocal microscope. The trajectories of the newly generated DGCs and their migrating distances were determined in Imaris. The angle of the cellular axis with respect to a line parallel to the GCL was measured using the angle tool in Image J. Confocal images of dendritic trees were analyzed in ImageJ using the NeuronJ plugin.

**Statistical analysis**. Data were analyzed with independent-samples Student's t tests, one-way, and two-way analyses of variance (ANOVAs) followed by least-significant difference tests, one-way ANOVAs with repeated measures followed by the Pearson correlation analysis, and the Kruskal–Wallis test. Two-tailed P values of < 0.05 were considered the cutoff for statistical significance. All data are represented as the means ± standard errors. Samples sizes (n values) represent the numbers of animals unless otherwise specified.

**Reporting summary**. Further information on research design is available in the Nature Research Reporting Summary linked to this article.

## Data availability
The datasets generated during and/or analyzed during the current study are available from the corresponding author on reasonable request.

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

## Acknowledgements

We thank Drs. Joel Levine, Carlos Lois, Maya Shelly, and Rachel Kery for their critical feedback on this manuscript. We thank Dr. Songhai Shi for providing CX43-related DNA vectors. This work was supported by the National Institutes of Health (grants NS089770, AG046875, NS104868 to S.G. and 1F30MH110103 to G.W.K.) and American Heart Association (18PRE34080158 to J.S.).

## Author contributions

J.W. and S.G. conceived and designed experiments. J.W., J.S., G.W.K. and Y.G. carried out experiments and analyzed data. S.J. provided support for in vitro time lapse imaging. All authors have participated in writing or discussing the manuscript.

## Additional information

**Competing interests:** The authors declare no competing interests.

