## [Peer Review File · Nature Communications]

Reviewers' comments:

Reviewer #1 (Remarks to the Author):

In this manuscript, Wang et al used in vivo endoscopic imaging to examine the dynamics of lateral dispersion of newborn progeny in the adult dentate gyrus which was found to occur at 5 days after retroviral labeling. Furthermore, they showed that physically adjacent newborn progeny are electrically coupled, and disruption of electrical coupling by CX43 deletion or expressing dominant negative Cx43 in newborn progeny impairs lateral dispersion of newborn progeny. Finally, they showed that lateral dispersion is required for subsequent circuit integration of the newborn neurons. This is an exciting pioneering study that affords a unique perspective into neurogenesis in the adult dentate gyrus in vivo. I have a few concerns summarized below that should be addressed before publication.

Major points:

1. Endoscopic imaging is a powerful technique that allows the in vivo visualization of the dynamics of stem cells and their progeny, an important next step in the field of adult neurogenesis and adult neural stem cell biology. However, this technique requires aspiration of some cortical tissues, does the injury from aspiration of cortical tissues damage the neurogenic niche and affect migration dynamics? The authors should control for this by analyzing sections of an untreated mouse for numbers, morphology, numbers. Post-hoc examination of the tissues after imaging will also help address this issue.
2. Better characterization of the populations at the different time points should be presented. Co-immunostaining should be performed to identify specifically which stage of the lineage is being imaged.
3. It has been suggested by a recent study (Sun et al, PNAS 2015, "Tangential migration of neuronal precursors of glutamatergic neurons in the adult mammalian brain") that newborn progeny show tangential migration along blood vessels. Do the authors observe migration along blood vessels in vivo? Are cells always associated with vessels? Is that possible that blood vessels serve as an intermedator to control the dispersion, since they are also express Cx43, thus newborn progeny and blood vessels could be electrically coupled and disruption of gap junctions in each of these components could also affect dispersion? These possibilities should be examined and discussed.
4. Authors did not observe the cell death of the newborn progeny from 5-7 dpi, does that mean cell death happened before those time points, since newborn progeny undergo significant cell death during the progenitor stage (Sierra et al., 2011 Cell Stem Cell)? Therefore, it is necessary to examine the cell stages associated with those time points.

Minor points:

1. It would be more informative that authors could comment on the directionality of the newborn progeny migration.
2. In Figure 2, why the response tom the recording cell is smaller than the response from the cell under stimulation if they are electrically coupled?
3. Do the cells electrically coupled derive from the same mother cell? Alternatively, distance matters most so that as long as they are physically close then they could be electrically coupled?
4. Does dnCx43 affect proliferation of the newborn progeny?

5. Do the authors expect circuit integration of newborn progeny be equally affected by dnCx43 and Cx43 deletion?

Reviewer #2 (Remarks to the Author):

The manuscript by Wang and collaborators studies the mechanisms involved in the lateral dispersion of adult born DGCs along the subgranular zone (SGZ) in the dentate gyrus of the mouse hippocampus. They also aim to establish whether such lateral dispersion is relevant to the subsequent circuit integration. By using the novel miniature endoscope live imaging technique, the authors analyze migration dynamics of newly generated DGCs in freely moving mice. They show that horizontal dispersion along the SGZ occurs in neurons at around 5 to 7 days of age (days post infection or dpi). Neurons migrate in small groups in a coordinated fashion, and sometimes they alternate the leading position, a process the authors refer to as "leapfrog" like, as one neuron bypasses the leading one and takes the new lead in the migration (this occurs in 35% of the cases). These results were corroborated in adult brain organotypic sections. The authors also showed that, at 5 dpi, some cells are coupled via gap junctions, and lateral dispersion is inhibited by genetic deletion of connexin 43 that constitutes gap junction hemichannels. When lateral dispersion was prevented, defects in morphology were observed in 14-dpi DGCs. The authors concluded that lateral dispersion of newly generated DGCs in the neurogenic zone is required for circuit integration.

Experiments are well performed and the authors use elegant approaches such as endoscope imaging to track the initial developing days of DGCs. Then most experiments are continued in slice cultures. The manuscript is well organized, but there are major caveats that need to be addressed.

1. There is no direct evidence on the relevance of lateral dispersion for neuronal development.

2. Electrical coupling between 5-7 dpi neurons was observed in only 4 out of 27 pairs in electrophysiological recordings, which brings a major concern regarding the incidence (less than 15 %) and the relevance of this phenomenon. In addition, Kunze et al 2009 showed that gap junctions between radial glia-like cells in the SGZ are required for neurogenesis in the adult dentate gyrus, but found that 50-75% of cells were electrically coupled. In this context, the 15% coupling observed here might be a vestige of the initial coupling that characterizes RG-like cells.

3. Since electrical coupling was present in a small proportion of new neurons, it is difficult to establish causality between lateral dispersion and the effect observed in the morphology of 14 dpi neurons. Moreover, in those experiments, the authors analyzed morphology in cells whose connexin 43 (CN43) function has been impaired by the expression of a dominant negative protein starting at the neural stem cell stage. It is unclear whether altered morphology was due to lack of lateral migration or other alterations that may be direct consequence of CN43 impairment at early neuronal differentiation (Kunze et al 2009). An alternative method to prevent lateral migration should be designed to actually link migration and neuronal integration.

4. Shorter dendrites are interpreted as impaired functional integration in 14 dpi neurons lacking CN43. However, functional integration of glutamatergic inputs barely starts by 14 dpi. To establish whether integration is altered would require studying more mature neurons and, preferably, through electrophysiological recordings.

2. *Better characterization of the populations at the different time points should be presented. Co-immunostaining should be performed to identify specifically which stage of the lineage is being imaged.*

The reviewer suggested to identify the lineage of cells that had been analyzed.

In response to this suggestion, we performed staining of the GFP+ cells at 1 and 5 days post viral infusion with GFAP, MCM2 and Prox1, showing that most labeled cells were MCM+ and GFAP+ at 1dpi. At 5dpi, we barely detected MCM signal in GFP+ cells, but most were Prox1 positive, suggesting these cells are postmitotic. Together with the morphological evidence in **Figure 1**, these GFP+ cells at 5dpi represent a population of immature dentate granule cells. We presented this additional data in **Supplementary Figure 1**.

3. *It has been suggested by a recent study (Sun et al, PNAS 2015, "Tangential migration of neuronal precursors of glutamatergic neurons in the adult mammalian brain") that newborn progeny show tangential migration along blood vessels. Do the authors observe migration along blood vessels in vivo? Are cells always associated with vessels? Is that possible that blood vessels serve as an intermedator to control the dispersion, since they are also express Cx43, thus newborn progeny and blood vessels could be electrically coupled and disruption of gap junctions in each of these components could also affect dispersion? These possibilities should be examined and discussed.*

The reviewer asked whether it is possible that the microvasculature serves as an intermedator to control the lateral dispersion of newborn neurons. In response to this question, we reanalyzed the *in vivo* images and discussed this interesting point.

As pointed out by the reviewer, a recent study using transgenic labeling showed that newborn neurons exhibited an association with microvessels. Following the suggestion by the reviewer, we further analyzed our videos. Among 11 cells, we indeed found 2 cells exhibited an impressive migration along a microvessel: wrapping and releasing and wrapping, then eventually moving forward, as shown in **Supplementary Video 3**. This suggests the possibility of a vessel-based migration mechanism. However, we did not observe a clear association between migration and microvessels for most newborn DGCs, which instead exhibited a pattern of typical interkinetic nuclear migration (**Supplementary Video 2-3**). We thus suspected that the 'leapfrog' dispersion might be another important mechanism by which newly generated DGCs disperse in the neurogenic zone. Importantly, in this study, we focused on only analyzing newborn DGCs between 5 dpi and 8 dpi, raising the possibility that the dispersion may employ different mechanisms across the initial development. The very young newly generated DGCs might show higher association with the microvessels, which requires further analysis. Inspired by the reviewer, we included a short paragraph to discuss this point.

In this study, we interfered with CX43 via retroviral targeting of newborn neurons. Another cell population, astrocytes that wrap vessels, also expressing CX43, but interestingly, as shown previously, newborn DGCs exhibit little coupling with these astrocytes (Kunze et al., 2009). This suggests that our CX43 manipulation was likely restricted to newborn neurons.

4. *Authors did not observe the cell death of the newborn progeny from 5-7 dpi, does that mean cell death happened before those time points, since newborn progeny undergo significant cell death during the progenitor stage (Sierra et al., 2011 Cell Stem Cell)? Therefore, it is necessary to examine the cell stages associated with those time points.*

The reviewer raised an interesting point. As suggested above, we have performed and presented GFAP, MCM and Prox 1 staining as shown in **Supplementary Figure 1**, and PCNA and DCX staining in **Supplementary Figure 5b**. These additional experiments showed that GFP expressing newborn neurons at 1 dpi were in the proliferative state, and we assume the robust cell death occurs during this phase as the reviewer pointed out and referenced (Sierra et al., 2011) and previously described (Ge et al., 2006). At 5dpi, GFP expressing cells largely represent immature DGCs (**Supplementary Figure 1** and **Supplementary Figure 5b**).

Minor points:

1. *It would be more informative that authors could comment on the directionality of the newborn progeny migration.*
This is a great point. A large portion of cells migrated in certain direction, but we indeed observed a portion migrated back and forth including the one we found wrapping vessel (**Supplementary Video 3**). This seems to be a critical phase, during which newborn neurons are competitively seeking for an opportunity for integration. We included a short description in the main text to present this interesting phenomenon.

2. *In Figure 2, why the response from the recording cell is smaller than the response from the cell under stimulation if they are electrically coupled?*

A given neuron may couple with several others, which may shunt the initial stimulation. We added a short description in the result to present this phenomenon.

3. *Do the cells electrically coupled derive from the same mother cell? Alternatively, distance matters most so that as long as they are physically close then they could be electrically coupled?*

This is a great question. We have recorded newborn neurons totally based on their physical distance. The coupling efficiency has been relatively low, suggesting distance is not of essence. In an additional set of experiments (**Supplementary Fig. 5**), we presented the data we recorded from cell clusters at 5 dpi (likely derived from a single progenitor). The efficiency was as high as 50% of recorded pairs. Considering that we recorded from acutely-prepared brain sections which likely underestimates the efficiency, the coupling *in vivo* may be even higher. Therefore, the cells derived from a single progenitor appear preferentially coupled. We have included this conclusion in the main text.

4. *Does dnCx43 affect proliferation of the newborn progeny?*

Per Kunze et al. (Kunze et al., 2009), if depletion occurred in RGL cells there was a strong effect on the proliferation. We have not observed an obvious reduction in the number of newborn DGCs if we induced a depletion from day 1 till day 3 after viral injection. This may be because retrovirus preferentially target transient amplifying cells. To verify our observations, we have induced CX43 manipulation after 4 days post viral injection (**Fig. 4d, f-j** and **Supplementary Fig. 7**). We found similar defects as those in Fig. 4a-b. However, we have revisited all the text to make sure of this 'inducible' verification.

5. *Do the authors expect circuit integration of newborn progeny be equally affected by dnCx43 and Cx43 deletion?*

speeded lateral dispersion, which might contribute to the EE-enhanced integration of newly generated DGCs (**Supplementary Fig. R2**).

In addition to these newly included experiments, we referred to several elegant papers in sections of introduction showing the importance of dispersion in the neurogenic zone in proceeding circuit formation in embryonic brain development (Wash et al, 2001, and Tsui et al. 2007). Altogether, in the revised manuscript, we feel that the justification of the role of lateral dispersion in regulating circuit formation has been substantially strengthened. We thought the dispersion of newborn DGCs in the neurogenic zone provides a prerequisite so that these horizontally-positioned cells could find a right place to initiate their circuit integration into the existing neural circuit.

2. Electrical coupling between 5-7 dpi neurons was observed in only 4 out of 27 pairs in electrophysiological recordings, which brings a major concern regarding the incidence (less than 15 %) and the relevance of this phenomenon. In addition, Kunze et al 2009 showed that gap junctions between radial glia-like cells in the SGZ are required for neurogenesis in the adult dentate gyrus, but found that 50-75% of cells were electrically coupled. In this context, the 15% coupling observed here might be a vestige of the initial coupling that characterizes RG-like cells.

The reviewer raised this concern in two aspects: 1) why the coupling ratio is so low, which affects the appreciation of physiological significance; 2) whether the coupling between newborn neurons are a vestige of initial coupling of RG-like cells as reported in the paper of Kunze et al. (2009). In response to this concern, we performed a few sets of additional experiments.

1) In the previous submission, we recorded the electrical coupling of newborn neurons using a common GFP retrovirus. Although during preparing brain sections we may have disrupted some couplings as discussed in a previous study (Yu et al., 2012), introducing an underestimation of coupling ratio, we agree with the reviewer that the ratio was low. During embryonic cortical development, ontogenetically-related neurons have been found to be tightly electrically coupled (Yu et al., 2012). Upon this concern, as shown in **Supplementary Fig. 5a-b** and **Fig. 3**, we used the newly developed method to sparsely label newborn DGC clusters, largely representing neurons from a single progenitor (**Fig. 3**, and (Kirschen et al., 2017)). We found that 5 out of 10 tested pairs were electrically coupled at 5 dpi. Given that the recording was performed on brain sections, which underestimates the coupling efficiency due to the nature of the sectioning procedure, we expect an even higher coupling efficiency *in vivo*. Consistent with our previous observation (**Fig. 2**), at 7dpi we did not see any coupling (**Supplementary Fig. 5d**).

2) As raised by the reviewer, whether the coupling is a vestige of the coupling between RGL cells. This is a very interesting point. RGL cells have been shown to be highly coupled, which has been found to be important to the proliferation and differentiation of neural progenitors (Kunze et al., 2009). In a pilot study, we therefore tested whether RGL cells and newborn DGCs are coupled. We labeled RGL cells with the injection of lenti-GFAP-Cre-GFP. Newborn DGCs were birth-dated and labeled with Flex-dTomato. We recorded 5 pairs between RGL cell and newborn neuron at 5 dpi, and did not see any functional coupling, confirming the finding that reported in the study by Kunze et al. (2009) no coupling

between RGL cell and new neuron. This suggests little overlapping between these two types of couplings. We included a short discussion on this interesting point.

3. Since electrical coupling was present in a small proportion of new neurons, it is difficult to establish causality between lateral dispersion and the effect observed in the morphology of 14 dpi neurons. Moreover, in those experiments, the authors analyzed morphology in cells whose connexin 43 (CN43) function has been impaired by the expression of a dominant negative protein starting at the neural stem cell stage. It is unclear whether altered morphology was due to lack of lateral migration or other alterations that may be direct consequence of CN43 impairment at early neuronal differentiation (Kunze et al 2009). An alternative method to prevent lateral migration should be designed to actually link migration and neuronal integration.

The reviewer is concerned about 1) coupling in only a small percentage of newborn neurons; 2) expression of CX43 at neural stem cell stage. Therefore, the question is whether this finding was due to any indirect effect. In addition, the reviewer recommended that an alternative method to regulate lateral dispersion should be designed. In response to this concern, we performed additional experiments and edited the main text accordingly.

1) In the response to question 2, we summarized that we have recorded cell clusters, likely from a single neural progenitor, at 5 and 7 dpi (**Supplementary Fig. 5**). The cells in these individual clusters showed higher coupling efficiency.

2) We understand the concern on potential effects on neural stem cells. In additional tests, we used two strategies to manipulate the expression of CX43 avoiding any confound effects on the proliferation and differentiation of stem cells: a) the Cre recombinase expression was induced in newborn neurons at 4 days after retroviral labeling by using doxycycline-inducible PTet-on vector (**Supplementary Fig. 7**) (Kumamoto et al., 2012; Rao et al., 2018); b) the expression of dnCX43 was induced to express in newborn neurons from 4 days after retroviral labeling of newborn neurons using the same vector (**Fig. 4d, f, g, h, i and j**). We found that the size of cell clusters was similar between control and CX43 manipulation groups, and similar to those in **Fig. 3**. More importantly, the integration defects at 14 and 21 were observed when we induced the expression of transgene (Cre with floxed CX43 mice or dnCX43) in newborn DGCs at 4 dpi. We have presented this set of results for excluding this concern.

3) As suggested, we performed another cluster of additional experiments in which we trained mice from 5 days after retroviral labeling to 8 dpi under enriched environment (EE). As shown in **Supplementary Fig. R2a-c**, we found that EE training sped up lateral dispersion between 5 and 8. This sped up lateral dispersion appeared to facilitate the integration of newborn neurons analyzed at 14 dpi. When we depleted CX43 from newborn DGCs at 4 dpi, we found that the EE-facilitated lateral dispersion was nearly abolished (**Supplementary Fig. R2d-f**). More importantly, we found that EE facilitated integration was also nearly abolished at 14 dpi. Together with the evidence showing **Figure 4e**, in which an induction of dnCX43 at -7-8 dpi in newborn neurons exhibited little effect on their integration, this data helps us understand the role of lateral dispersion in regulating integration.

4. Shorter dendrites are interpreted as impaired functional integration in 14 dpi neurons lacking CN43. However, functional integration of glutamatergic inputs barely starts by 14 dpi. To establish whether integration is altered would require studying more mature neurons and, preferably, through electrophysiological recordings.

The reviewer recommended to analyze functional integration of newborn neurons at a later integration stage besides the onset phase of synapse formation. As suggested by the reviewer, we performed four additional groups of experiments at 21 dpi, at which age newborn neurons are expected to be well integrated under physiological condition.

In the first set of experiments (as presented in **Figure 4f, g, h, i and j**), we induced the expression of GFP only or dnCX43 tagged with GFP in retrovirus-labelled cells at 4 days after viral infusion with doxycycline (DOX) as those in **Figure 4d** or **previously described** (Rao et al., 2018). We analyzed both morphological and functional integration at 21 dpi. We found that dnCX43+ cells exhibited shorter dendrites and fewer dendrite branch points. We also found that there was a sharp decrease in frequency of spontaneous excitatory postsynaptic currents. There was no detectable change in the amplitudes of these spontaneous responses, suggesting fewer synapses have been formed.

To confirm this observation, in the second set of tests (**Supplementary Figure 7a, b and c**), we used the floxed CX43 mice by labeling newborn neurons with an inducible retrovirus carrying Cre recombinase gene. We induced Cre expression to deplete the CX43 at 4-5dpi (2 days DOX). We performed similar experiments as those in **Figure 4g** and **h**. As expected, there were very severe integration defects in these CX43-null newborn DGCs.

We also performed a third set of tests, in which we induced the Cre recombinase expression at 8-9 dpi (2 days DOX). The dendritic length and branch points were comparable to those cells expressing GFP only. Given the fact of similar data collection as those in **Figure 4e**, we did not include a supplementary figure for this result.

References

- Ge, S., Goh, E.L., Sailor, K.A., Kitabatake, Y., Ming, G.L., and Song, H. (2006). GABA regulates synaptic integration of newly generated neurons in the adult brain. *Nature* **439**, 589-593.
- Kirschen, G.W., Shen, J., Tian, M., Schroeder, B., Wang, J., Man, G., Wu, S., and Ge, S. (2017). Active Dentate Granule Cells Encode Experience to Promote the Addition of Adult-Born Hippocampal Neurons. *The Journal of Neuroscience* **37**, 4661.
- Kumamoto, N., Gu, Y., Wang, J., Janoschka, S., Takemaru, K.-I., Levine, J., and Ge, S. (2012). A role for primary cilia in glutamatergic synaptic integration of adult-born neurons. *Nat Neurosci* **15**, 399-405.
- Kunze, A., Congreso, M.R., Hartmann, C., Wallraff-Beck, A., Huttmann, K., Bedner, P., Requardt, R., Seifert, G., Redecker, C., Willecke, K., *et al.* (2009). Connexin expression by radial glia-like cells is required for neurogenesis in the adult dentate gyrus. *Proc Natl Acad Sci U S A* **106**, 11336-11341.
- Rao, S., Kirschen, G.W., Szczerkowska, J., Di Antonio, A., Wang, J., Ge, S., and Shelly, M. (2018). Repositioning of Somatic Golgi Apparatus Is Essential for the Dendritic Establishment of Adult-Born Hippocampal Neurons. *J Neurosci* **38**, 631-647.
- Sun, G.J., Zhou, Y., Stadel, R.P., Moss, J., Yong, J.H., Ito, S., Kawasaki, N.K., Phan, A.T., Oh, J.H., Modak, N., *et al.* (2015). Tangential migration of neuronal precursors of glutamatergic neurons in the adult mammalian brain. *Proc Natl Acad Sci U S A* **112**, 9484-9489.

Yu, Y.C., He, S., Chen, S., Fu, Y., Brown, K.N., Yao, X.H., Ma, J., Gao, K.P., Sosinsky, G.E., Huang, K., *et al.* (2012). Preferential electrical coupling regulates neocortical lineage-dependent microcircuit assembly. *Nature* 486, 113-117.

Supplementary Figure for reviewer 2

Supplementary Figure R2 Environmental enrichment accelerated lateral dispersion and integration of newborn DGCs

a) Shown is the experimental timeline of ontogenetic birth date labeling and subsequent exposure to standard (Std) or enriched (Enr) environments.

b) Shown are representative images of ontogenetically-related cell clusters at 6 and 8 dpi from Std and Enr groups. The scale bar is 20 μm.

c) Shown is a plot of the average nearest neighbor distances between ontogenetically-related daughter cells at 6, 8, and 14 dpi of mice housed in Std and Enr environments. Two-tailed unpaired t-tests, $P = 0.0057$, $P > 0.05$, $P > 0.05$, respectively.

d) Shown is a table of electrophysiological properties of 6 and 8 day old ontogenetically-labeled DGCs of mice housed in Std or Enr environments. Std: 6 cells from 2 mice; Enr: 8 cells from 3 mice.

e) Shown is the experimental timeline of CX43 deletion from newborn DGCs with housing in enriched or standard environments.

f) Shown is a plot of nearest neighbor distance from 6-day-old DGCs from GFP control and CX43-deleted conditions of mice housed in Std or Enr environments. Two-tailed unpaired t-tests, $P = 0.0082$, $P > 0.05$, respectively.

g) Shown is a plot of nearest neighbor distance from 8-day-old DGCs from GFP control and CX43-deleted conditions of mice housed in Std or Enr environments. Two-tailed unpaired t-tests, $P > 0.05$, $P > 0.05$, respectively.

h-i) Shown is a plot of number of branch points of GFP control and CX43-deleted 14-day-old DGCs of mice housed in standard or enriched environments. Two-tailed unpaired t-tests, $P = 0.007$, $P > 0.05$, respectively.

REVIEWERS' COMMENTS:

Reviewer #1 (Remarks to the Author):

The authors have fully addressed my concerns and now the manuscript is ready to be published as its current format.

Reviewer #2 (Remarks to the Author):

The manuscript has been largely improved. In particular, new experiments showing a higher coupling efficiency (5 out of 10 pairs recorded in Supplementary Fig. 5) make the main argument more relevant.

As it stands, data shown in Supp Fig. 5 makes a stronger argument than data shown in main Figure 2, where coupling is only about 20%. The authors should include the new data shown in this Supp figure in the main body of the manuscript (perhaps send data from Fig. 2 to supplementary information), add the electrophysiological recordings that support the bar charts, and adequately state the sample size (number of mice) used to collect those 10 paired recordings. After that change is done, the manuscript should be suitable for publication.

Response to the reviewer 2's remaining comments:

Reviewer #1 (Remarks to the Author):

The authors have fully addressed my concerns and now the manuscript is ready to be published as its current format.

Reviewer #2 (Remarks to the Author):

The manuscript has been largely improved. In particular, new experiments showing a higher coupling efficiency (5 out of 10 pairs recorded in Supplementary Fig. 5) make the main argument more relevant.

As it stands, data shown in Supp Fig. 5 makes a stronger argument than data shown in main Figure 2, where coupling is only about 20%. The authors should include the new data shown in this Supp figure in the main body of the manuscript (perhaps send data from Fig. 2 to supplementary information), add the electrophysiological recordings that support the bar charts, and adequately state the sample size (number of mice) used to collect those 10 paired recordings. After that change is done, the manuscript should be suitable for publication.

We thank this suggestion. We have merged the Supp Fig. 5 to the Fig. 2, and included the sample size. Both the text and legend have been revised accordingly.